# Lattice Distortion and Phase Stability of Pd-Doped NiCoFeCr Solid-Solution Alloys

**DOI:** 10.3390/e20120900

**Published:** 2018-11-25

**Authors:** Fuxiang Zhang, Yang Tong, Ke Jin, Hongbin Bei, William J. Weber, Yanwen Zhang

**Affiliations:** 1Materials Science and Technology Division, Oak Ridge National Laboratory, 1 Bethel Valley Rd, Oak Ridge, TN 37831, USA; 2Department of Materials Science and Engineering, University of Tennessee, Knoxville, TN 37996, USA

**Keywords:** solid-solution alloys, lattice distortion, phase transformation

## Abstract

In the present study, we have revealed that (NiCoFeCr)_100−*x*_Pd*_x_* (*x*= 1, 3, 5, 20 atom%) high-entropy alloys (HEAs) have both local- and long-range lattice distortions by utilizing X-ray total scattering, X-ray diffraction, and extended X-ray absorption fine structure methods. The local lattice distortion determined by the lattice constant difference between the local and average structures was found to be proportional to the Pd content. A small amount of Pd-doping (1 atom%) yields long-range lattice distortion, which is demonstrated by a larger (200) lattice plane spacing than the expected value from an average structure, however, the degree of long-range lattice distortion is not sensitive to the Pd concentration. The structural stability of these distorted HEAs under high-pressure was also examined. The experimental results indicate that doping with a small amount of Pd significantly enhances the stability of the fcc phase by increasing the fcc-to-hcp transformation pressure from ~13.0 GPa in NiCoFeCr to 20–26 GPa in the Pd-doped HEAs and NiCoFeCrPd maintains its fcc lattice up to 74 GPa, the maximum pressure that the current experiments have reached.

## 1. Introduction

High-entropy alloys (HEAs) are usually a single-phase solid-solution with multi principle elements randomly distributed in the lattice [1,2]. Due to the size difference of individual atoms, lattice distortion is believed to be one of the core effects, which greatly affects the mechanical and physical properties [3,4,5,6,7,8]. The distorted local lattice provides pinning sites to slow down dislocation motion and therefore improve the mechanical performance of high-entropy alloys [9,10,11,12]. The intrinsic lattice distortion in HEAs can shorten the free-electron migration paths and reduce the electrical and thermal conductivities [1,13,14], which can enhance the recombination of radiation defects due to a strong localized heating effect. In addition, distorted local lattice sites can retard the motion of radiation defects to delay their growth. Therefore, HEAs are strong candidates for nuclear materials by showing excellent radiation resistance [13,15]. However, a quantitative description of the lattice distortion in HEAs is a challenge and previous experimental results are controversial [1,16,17]; especially with respect to alloys with the fcc structure. For example, no obvious lattice distortion was reported previously for a NiCoFeMnCr alloy [17]. Recently, we developed a new analytical method based on atomic pair distribution function (PDF) measurement that can quantitatively describe the local lattice distortion in some high- and medium-entropy alloys [18,19,20]. PDF analysis has shown that the local lattice distortion in the NiCoFeCr is negligible (< 0.1%), while the NiCoFeCrPd HEA has a very large local lattice distortion (0.79%). Since there is a large mismatch of atomic size between Pd and other atoms, it is interesting to investigate the effect of Pd content on the lattice distortion in (NiCoFeCr)_100−*x*_Pd*_x_* solid-solution alloys.

Besides atomic size mismatch, different atomic configurations can also affect local lattice distortion in HEAs. For instance, the local bonding environment of individual atoms varies in solid-solution alloys, leading to the fluctuation of nearest atomic pair distances and short-range order [21]. Extended X-ray absorption fine structure (EXAFS) is atomic mass sensitive and is a powerful tool to measure the distance of different atomic pairs in solid-solution alloys. However, some approximations need to be made for those alloys with components that are neighbors in the periodic table. With EXAFS measurement, we have successfully revealed the short-range order in NiPd [22] and NiCoCr [21] solid-solution alloys. 

The uncertainty for atoms being located exactly on the lattice sites is another type of structural disorder that contributes to the excess configurational entropy [1,23]. Previous TEM analysis indicated that the lattice distortion destabilized the structure, and phase segregation was observed in solid-solution alloys, such as NiCoFeCrMn [24,25], during annealing and ion irradiation. Under high pressure, the fcc lattice of some alloys can transform to another close-packed hcp structure. However, the phase transformation behavior in the HEAs is not simple. Experiments demonstrated that NiCoFeCr [26] and NiCoFeCrMn [27,28] started to transform to an hcp structure at ~13 GPa, whereas phase transition was not found in NiCoFeCrPd alloy even up to 74 GPa [26]. Moreover, it recognized that magnetic contributions to the free energy may play a critical role. In this paper, the long/short-range lattice distortion and structural stability of Pd-doped (NiCoFeCr)_100−*x*_Pd_*x*_ HEAs were experimentally studied with total X-ray scattering, X-ray diffraction methods and the local bonding environment of atoms in the solid-solution is derived by EXAFS measurements.

## 2. Materials and Methods 

Elemental metals Ni, Co, Fe, Cr, and Pd (> 99.9% pure) in the designed atomic ratios with the formula of (NiCoFeCr)_100−*x*_Pd*_x_* (*x* = 1, 3, 5 and 20) were carefully weighed and mixed by arc melting. The arc-melted buttons were flipped and re-melted at least five times before drop casting to ensure the homogeneity. The ambient total scattering measurements were performed at synchrotron beamline F2 of CHESS (Cornell High Energy Synchrotron Source), Cornell University, with an X-ray energy of E = 61.332 KeV and beam size of 500 × 500 μm^2^. A two-dimensional stationary detector with 200 × 200 μm^2^ pixel size was placed ∼20 cm behind the sample to collect data. Fit2D software [29] was used to correct for a beam polarization and a dark current. In order to obtain real-space PDF, the measured patterns were Fourier transformed by PDFgetX3 [30] and then normalized reciprocal-space structure function in a Q range of 30 Å^−1^. Using PDFGui software [31], the measured PDFs were refined with the fcc structure models. The in situ high-pressure XRD (X-ray powder diffraction) was conducted with the diamond anvil cell technique in transmission mode at beamline 16-BM-D of the APS (Advanced Photon Source), Argonne National Laboratory. For high-pressure XRD experiments, a methanol/ethanol (4/1) mixture was used as the pressure transition medium. The wavelength of the X-ray was 0.4989 Å and 0.3103 Å for ambient and high-pressure measurements, respectively. For all of the synchrotron XRD experiments, the instrument parameters were calibrated with CeO_2_ as the standard, and XRD profiles were analyzed with the Rietveld refinement method using the program Fullprof [32]. The EXAFS spectra at the K-edge of elements Ni, Co, Fe, Cr were conducted in a fluorescence mode with a grazing exit configuration (grazing angle of 2–3°) at beam 13-ID-E of APS, Argonne National Laboratory. Athena program [33] was used for the reduction and analysis of the EXAFS data. The fitting of the EXAFS spectra was conducted with Artemis in Demeter software package [33] in a fixed k-range (3.0–12.0 Å^−1^) and an fcc structural model was used to simulate the structure of the solid solutions.

## 3. Results and Discussion

### 3.1. Local Lattice Distortion

The local lattice distortion induced by atomic size mismatch in HEAs has been estimated by a hard sphere model [34]. However, an experimental investigation has revealed that the hard-sphere model considerably overestimated the local lattice distortion in the HEAs. The measurement of the Bragg peak width in XRD or neutron diffraction profiles contains information of both static and dynamic displacements, which, however, cannot be resolved [17]. PDF analysis based on total scattering measurements can effectively reveal the local lattice distortion in terms of variation of local bond distance. Figure 1a shows the observed PDF profile of the NiCoFeCrPd HEA and the calculated one based on a random solid-solution model. Except for the first peak, the calculated pattern matches the observed one very well. The mismatch in the first atomic shell is an indication of local lattice distortion in NiCoFeCrPd. In order to quantitively describe the local lattice distortion in solid-solution alloys, we introduced a local lattice distortion parameter *ε*.
(1)$$\varepsilon =\left(a_{var}-a_{avg}\right)/a_{avg}$$

Where *a_avg_* is the lattice parameter obtained from fitting the PDF profile over the whole r-range and *a_var_* is the lattice parameter obtained by fitting the PDF profile from r_min_ = 1.5 Å (data below this value was excluded because of large oscillations) to the varied r_max_ value. The local lattice distortion in the fcc HEAs is strongly localized in the first atomic shell, as shown in Figure 1a. Therefore, we only focused on the lattice strain in the first atomic shell, ε_1st_. Our results show that NiCoFeCr has a negligible ε_1st_ (< 0.1%), whereas the NiCoCr and FeCoNiCrMn possess a small positive ε_1st_, suggesting that the local bond distances are larger than the expected value from their average structures. NiCoFeCrPd has the largest ε_1st_ (0.79%) (Figure 1b) reported so far. The large lattice strain in NoCoFeCrPd is caused by the large size mismatch between Pd and other elements. With similar analysis of the Pd-doped (NiCoFeCr)_100−*x*_Pd*_x_* HEAs, the local lattice distortion is found to be proportional to the content of Pd (Figure 1b).

### 3.2. Long-Range Lattice Distortion

XRD experimental results have shown that the Bragg peaks are broadened and the intensities are reduced in HEAs [35] because of the larger uncertainty for atoms being exactly on the crystalline lattice sites. No obvious long-range lattice distortion has been observed previously from X-ray or neutron diffraction measurements. For most HEAs, their lattice remains the fcc, bcc or hcp structure. However, long-range lattice distortion is found in Pd-doped (NiCoFeCr)_100−*x*_Pd*_x_* HEAs from XRD measurement. Figure 2 shows the XRD patterns of (NiCoFeCr)_100−*x*_Pd*_x_* (*x* = 1, 3, 5 and 20 atom%) HEAs under ambient conditions. The red dots are observed patterns and the green line are calculated patterns based on Rietveld refinement. As shown in the enlarged patterns (Figure 2b), there is a clear deviation at the (200) Bragg peak. The observed (200) peak exhibits a larger d-spacing than the average, whereas all of the other Bragg peaks match the average positions very well. The deviation of the (200) lattice planes is ~0.004 Å. Since XRD reveals the long-range order, the mismatch of the (200) peak indicates that the Pd-doped NiCoFeCrPd*_x_* HEAs have a distorted lattice from the ideal fcc structure, though no peak splitting was observed. Experimental analysis also suggests that the deviation of the (200) peak in the alloys with different concentrations of Pd is nearly the same. Due to the substitution of larger Pd atoms into the structure, it is not difficult to understand the change of lattice constant (Table 1) and local lattice distortion with the Pd content in the solid-solution alloys. It is surprising that even 1atom% Pd-doping in (NiCoFeCr)_99_Pd_1_ HEAs can cause a long-range structural distortion on the (200) lattice planes. The systematic larger (200) lattice plane spacing suggests that the large Pd atoms may be not randomly distributed in the lattice. To obtain the short-range order information for these HEAs, a method capable of characterizing the local bonding environment is strongly needed.

### 3.3. Local Bonding Environment

Neither XRD nor total scattering measurements can give the atomic bonding information in the solid-solution alloys. In order to detect the bond distance in the (NiCoFeCr)_100−*x*_Pd*_x_* (*x* = 1, 3, 5 and 20 atom%) HEAs, we measured the K-edge X-ray absorption spectrums of Ni, Co, Fe, and Cr elements. Since these four elements have similar X-ray scattering ability, it is difficult to distinguish the individual elements in the solid-solution alloys. As an approximation, we assumed that, except for Pd, the atoms around the core have the same X-ray scattering ability. Figure 3 is the *k*^3^-weighted FTs of the Fe K-edge EXAFS for the (NiCoFeCr)_100−*x*_Pd*_x_* (*x* = 1, 3, 5 and 20 atom%) HEAs. We assumed that Pd and Fe atoms are randomly distributed in the fcc lattice and the red dash lines in Figure 3 are the fittings within the first shell in the radial distance of 1–3.5 Å. The first shell peak is generally well fitted with this approximation. The derived average distance between the nearest atomic pairs is shown in Table 1. In general, for the lower Pd-doped alloys, the distance measured with EXAFS is in good agreement with that measured with XRD but for the equiatomic NiCoFeCrPd alloy, the distance measured with EXAFS is obviously smaller (1.7%) than that from the XRD measurement, which suggests that there is a larger lattice strain in the NiCoFeCrPd HEA than the lower Pd-doped alloys. This is in agreement with the total scattering measurements. From the PDF analysis, we have confirmed that the local lattice distortion in NiCoFeCrPd is 3–4 times larger than that in the lower Pd-doped solid-solution alloys [20].

### 3.4. Structural Stability at High Pressures

Multi-component concentrated solid-solution alloys can have an fcc, bcc or hcp structure. Theoretical calculations suggested that some fcc HEAs are metastable because their hcp counterparts have similar Gibbs free energies at ambient conditions. Previous experiments have revealed that some of the fcc alloys can transform to the hcp structure under high-pressure conditions, such as NiCoFeCr [26] and NiCoFeCrMn [27,28] alloys that transformed to an hcp structure at ~13 GPa. The hcp structure is quenchable to ambient conditions, though the phase transition is very sluggish. The phase stability is composition sensitive. For FeMnCoCr alloys [36], the hcp structure can coexist with the fcc structure from the sample preparation process. By properly tuning the chemical composition, a dual-phase alloy can possess excellent mechanical properties. For the five-element system of NiCoFeCrPd, the fcc structure is stable up to 74 GPa [26]. The larger size of Pd atoms plays a key role in structural stability. It is thus interesting to study the effect of Pd content on the phase transition. We pressurized the Pd-doped (NiCoFeCr)_100−*x*_Pd*_x_* HEAs with diamond-anvil cell techniques, and the experimental results indicate that a small amount of Pd greatly affects the structural stability. The critical pressure for the fcc to hcp phase transition is strongly increased to more than 20 GPa in 1 atom%, 3 atom% and 5 atom% doped (NiCoFeCr)_100−x_Pd_x_ solid-solution alloys. Figure 4 shows the XRD profiles of 3 atom% Pd doped (NiCoFeCr)_97_Pd_3_ alloy at different pressures, and the hcp structure starts to appear at 20.9 GPa. The transition is very sluggish, and the amount of the hcp structure is only ~33% at 34.1 GPa. The hcp structure is stable once it is formed, and the quenched alloy has a mixed structure of fcc and hcp. The critical transition pressure is not sensitive to the amount of Pd doped and is observed at 26.0, 20.9, and 21.0 GPa for the 1%, 3% and 5% Pd-doped (NiCoFeCr)_100−*x*_Pd*_x_* HEAs, respectively. However, no hcp structure was found in the equiatomic solid-solution alloy NiCoFeCrPd up to 74 GPa [26]. Since Pd has a much larger atomic size than other elements in these alloys, the substitution of Pd for other atoms increases the lattice parameter and atomic-pair distances. When the Pd content in the alloys is sufficient, all the smaller atoms will have more free space to move, which may allow adaption to the lattice distortion at high pressures. This may be the main reason why Pd can cause changes in the critical transition pressure for Pd-doped NiCoFeCrPd solid-solution alloys. Therefore, the high local lattice distortion greatly in the equiatomic HEA can enhance the stability of the fcc lattice.

We further analyzed the long-range lattice distortion of (NiCoFeCr)_100−*x*_Pd*_x_* solid-solution alloys under high pressure, i.e., the deviation at the (200) Bragg peak. Figure 5 shows the deviation of each observed Bragg peaks at different pressures for the 1% Pd-doped NiCoFeCr. The deviation of the (200) Bragg peak obviously increased with pressure. Before the hcp structure starts to form, the deviation of (NiCoFeCr)_99_Pd_1_ has reached 0.01 Å. In a strict sense, the structure of the (NiCoFeCr)_99_Pd_1_ alloy is not fcc anymore. A similar behavior is also observed in the (NiCoFeCr)_97_Pd_3_ and (NiCoFeCr)_95_Pd_5_ alloys. Accordingly, the external high pressure can enhance the long-range lattice distortion.

Figure 6 shows the *P-V* curves of Pd-doped NiCoFeCr alloys. As a comparison, the *P*-*V* curves for NiCoFeCr and NiCoFeCrPd are also shown. When fitted with a 3rd Birch-Murnaghan equation of state, the bulk modulus is 190(5), 171(8) and 186(4) GPa for the 1%, 3% and 5% Pd-doped NiCoFeCr solid-solution alloys, respectively. The bulk modulus of the Pd-doped solid-solution alloys is smaller than that of NiCoFeCr (206 GPa) but larger than that of NiCoFeCrPd (168 GPa). The addition for larger Pd atoms makes the alloys more compressible with smaller bulk modulus because the addition of Pd atoms increased the lattice parameters and the smaller atoms, Ni, Co, Fe, and Cr, may have more “free” space to move in order to adapt the structure during pressurization.

## 4. Conclusions

We have systematically studied the effects of Pd doping in *(*NiCoFeCr*)_100−x_*Pd*_x_* solid-solution alloys on lattice distortion and phase stabilities. The short-range order is strongly dependent on the Pd content in the alloys. Both PDF and EXAFS measurements suggest that the lattice is locally strained in the highly Pd doped solid-solution alloys. Even a small amount of Pd addition in the alloys can cause a long-range lattice distortion by showing a larger (200) lattice plan spacing than the expected from the average structure. High-pressure studies revealed that a small amount of Pd in the solid-solution alloys greatly enhanced the phase stability of the fcc structure, and the critical pressure for the fcc to hcp transition increased from ~13 GPa to more than 20 GPa in 1%, 3% and 5% Pd-doped alloys, while the fcc is stable up to 74 GPa in the equiatomic NiCoFeCrPd HEA.

## Figures and Tables

**Figure 1 entropy-20-00900-f001:**
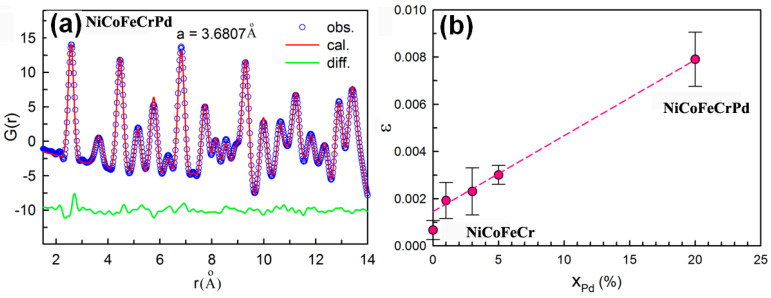
(**a**) Pair distribution function of NiCoFeCrPd HEA (high-entropy alloys). The blue symbols are experimental data and the red line is fit to the data using a random solid-solution model. The slight shift of the measured PDF (pair distribution function) from the fitted data (difference shown at *r* from 2 to 3 Å indicates the local lattice distortion; (**b**) Local lattice distortion in the first atomic shell as a function of the Pd concentration in the alloys.

**Figure 2 entropy-20-00900-f002:**
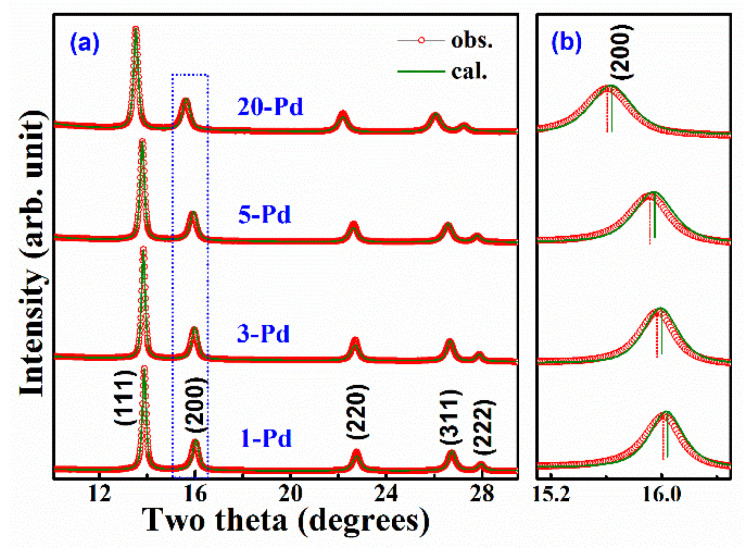
(**a**) The XRD (X-ray powder diffraction) profiles measured with synchrotron X-rays (*λ* = 0.4989Å). The red symbols are measured data and the green lines are calculated profiles based on Rietveld refinement; (**b**) the enlarged part of the XRD profiles and the observed (200) Bragg peak in all the samples obviously shifted to lower two theta angles with larger d-values.

**Figure 3 entropy-20-00900-f003:**
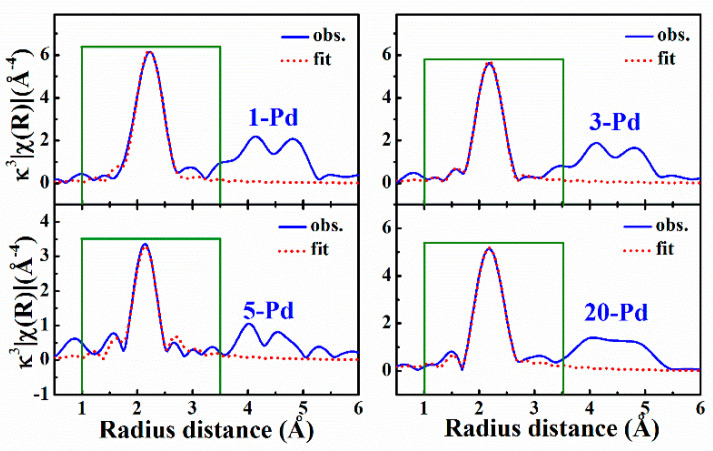
The *κ*^3^-weighted FTs (Fourier Transforms) of the Fe K-edge EXAFS (extended X-ray absorption fine structure) in (NiCoFeCr)_100−x_Pd_x_ solid-solution alloys. The solid blue line is observed and the red dash line is the fitting with the nearest neighbors.

**Figure 4 entropy-20-00900-f004:**
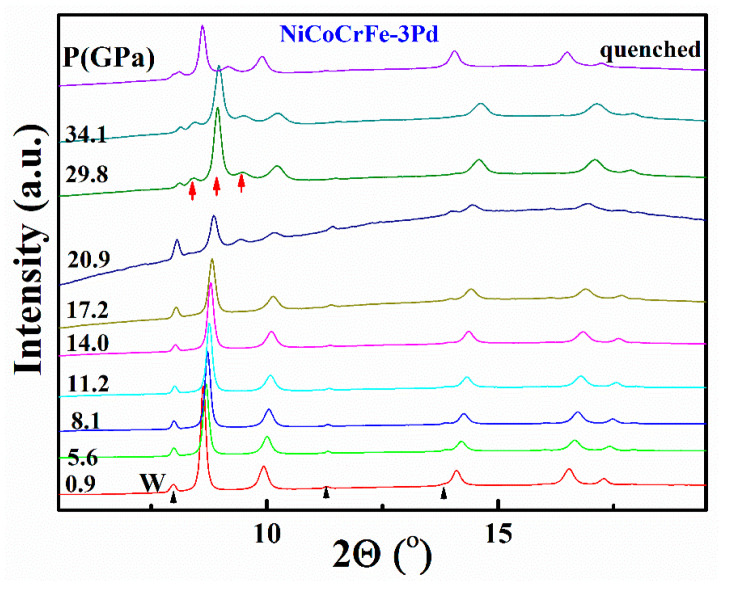
The XRD profiles of (NiCoFeCr)_97_Pd_3_ measured at different pressures. The fcc lattice starts to transform to hcp structure at 20.9 GPa. The weak diffraction peaks marked with small black arrows are from the W gasket during measurement.

**Figure 5 entropy-20-00900-f005:**
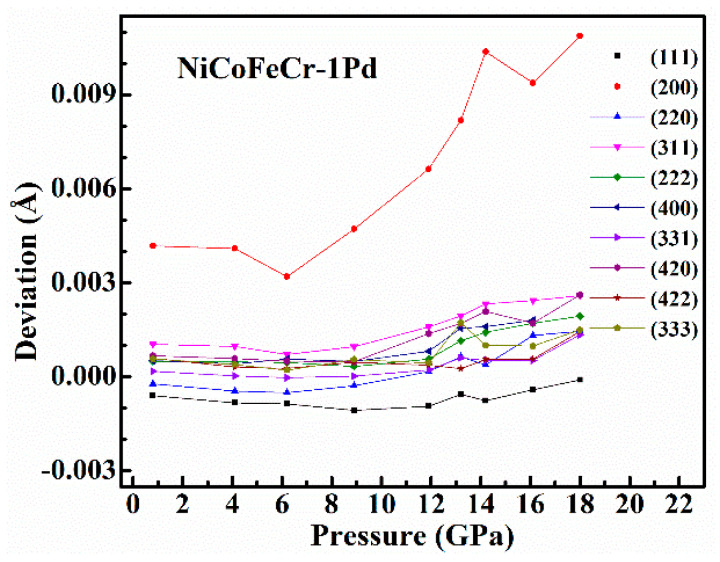
The deviation of the observed Bragg peaks from the ideal fcc structure at different pressures. The deviation of (200) peak is obvious and it increases with pressure.

**Figure 6 entropy-20-00900-f006:**
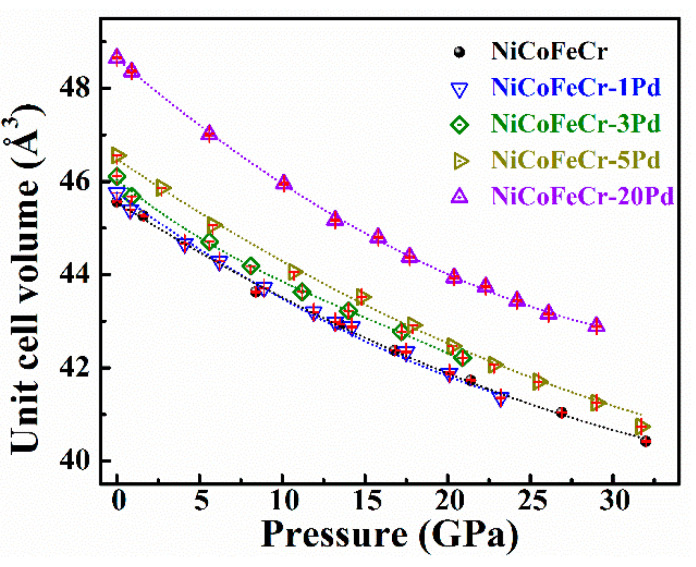
The pressure dependence of the unit cell volume. The dashed lines *P*-*V* curves which are fitted with 3-rd-order Birch-Murnaghan equation of state

**Table 1 entropy-20-00900-t001:** The lattice constant and the nearest atomic pair distance in the solid-solution alloys measured with XRD and EXAFS (extended X-ray absorption fine structure).

Sample	Lattice Constant (Å)	Nearest Atomic Pair Distance (Å)	
		XRD	EXAFS
(NiCoFeCr)_99_Pd_1_	3.5767(1)	2.5291	2.53(1)
(NiCoFeCr)_97_Pd_3_	3.5860(2)	2.5357	2.54(1)
(NiCoFeCr)_95_Pd_5_	3.5975(2)	2.5438	2.54(1)
NiCoFeCrPd	3.6679(4)	2.5936	2.55(4)

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
