# Peer review of "Lattice Distortion and Phase Stability of Pd-Doped NiCoFeCr Solid-Solution Alloys"

_entropy, 2018, doi:10.3390/e20120900_

Round 1
Reviewer 1 Report
This study is very interesting, and my comments are as follows:
1. Typing error, 1atm.% (Line 14)
2. The compositions of experimental alloys should be listed and examined to avoid simple calculation and weighting errors.
3. This manuscript did not show any micrographs of the alloys to prove that the alloys had a solid-solution single phase. Maybe the NiCoFeCrPdx alloys with higher Pd-content had the second phase or third phase.
4. The manuscript used XRD profiles to examine the fcc-to-hcp transformation, but the sensitivity of XRD is not good enough to detect the phase with small volume fraction. Maybe the transformation occurs at lower pressure. The authors should consider to examine this transformation by TEM.
Author Response
We appreciate the careful check of our manuscript and the precious suggestions and comments from the reviewers and we have revised out manuscript accordingly. The detailed changes are highlighted in the revised version and all the response are listed below.
1. Typing error, 1atm.% (Line 14)
Corrected, thanks
2. The compositions of experimental alloys should be listed and examined to avoid simple calculation and weighting errors.
We have added the formula of the solid solution alloys in the Materials and Methods part
3. This manuscript did not show any micrographs of the alloys to prove that the alloys had a solid-solution single phase. Maybe the NiCoFeCrPdx alloys with higher Pd-content had the second phase or third phase.
Usually micrographs (eg. SEM picture) can show the microstructure of samples. In order to identify the phase structure, XRD measurements may be a better way. From the measured XRD profiles, all the samples show a single fcc structure and no other phases were observed.
4. The manuscript used XRD profiles to examine the fcc-to-hcp transformation, but the sensitivity of XRD is not good enough to detect the phase with small volume fraction. Maybe the transformation occurs at lower pressure. The authors should consider to examine this transformation by TEM.
The reviewer is right, the phase transition may happen at lower pressures if it can be observed with TEM, because the fcc-hcp transition for our samples is a very sluggish process. XRD can detect the new structure only the phase has certain quantity (such as 1-2wt%). However, TEM cannot can only observe the quenched sample. In our case, we measure all the samples in situ under similar conditions with XRD, and the observed phase transition and the critical transition pressures should be consistent and reliable.
Reviewer 2 Report
Dear authors,
This is a very nice work with impressive results. Congratulations. Although the subject is quite difficult for a reader to understand and follow up, the presentation of the state-of-the-art and the results is clearly and nicely done. The methodology, especially the atomic pair distribution function (PDF) deserves some more emphasis in the second section (materials and methods) according to my opinion. So do the softwares with the Greek names "Athena" and "Artemis". Please emphasise on the role of this analysis to the overall discussion presented in this work. It is quite impressive that even the small amounts of Pd additions cause long-range lattice distortion. Furthermore, I also find impressive that the addition of Pd atoms consists the alloys as more compressible, because as you write "the addition of Pd atoms increased the lattice parameters and the smaller atoms, Ni, Co, Fe, and Cr, may have more “free” space to move in order to adapt the structure during pressurization.".
Some minor typing mistakes exist in the manuscript, but I am sure you will correct them.
E.g. see lines 84 (contians), 85 (resloved), 97 (suggsting), 119 (revealss), 141 (elelemnts).
Very nice work.
Herewith, I accept its publication in its current form. You can please take into consideration to enhance somehow the methodology presented in section 2 according to my comments.
Sincerely,
The reviewer
Author Response
We appreciate the careful check of our manuscript and the precious suggestions and comments from the reviewers and we have revised out manuscript accordingly. The detailed changes are highlighted in the revised version and all the response are listed below.
This is a very nice work with impressive results. Congratulations. Although the subject is quite difficult for a reader to understand and follow up, the presentation of the state-of-the-art and the results is clearly and nicely done. The methodology, especially the atomic pair distribution function (PDF) deserves some more emphasis in the second section (materials and methods) according to my opinion. So do the softwares with the Greek names "Athena" and "Artemis". Please emphasise on the role of this analysis to the overall discussion presented in this work. It is quite impressive that even the small amounts of Pd additions cause long-range lattice distortion. Furthermore, I also find impressive that the addition of Pd atoms consists the alloys as more compressible, because as you write "the addition of Pd atoms increased the lattice parameters and the smaller atoms, Ni, Co, Fe, and Cr, may have more “free” space to move in order to adapt the structure during pressurization.".
We very appreciate the positive comments from the reviewer. We have improved our manuscript based on his suggestions. For the PDF method, we have added more details in the Materials and Methods part. For the EXAFS method, more details are also added in the same paragraph. In addition, we added some short descriptions for EXAFS method in the Introduction part and section 3.3.
Some minor typing mistakes exist in the manuscript, but I am sure you will correct them.
E.g. see lines 84 (contians), 85 (resloved), 97 (suggsting), 119 (revealss), 141 (elelemnts).
Thanks for finding these typos, and all have been corrected
Very nice work.
Herewith, I accept its publication in its current form. You can please take into consideration to enhance somehow the methodology presented in section 2 according to my comments.